# Genetic Diversity and Spatiotemporal Distribution of SARS-CoV-2 Variants in Guinea: A Meta-Analysis of Sequence Data (2020–2023)

**DOI:** 10.3390/v17020204

**Published:** 2025-01-31

**Authors:** Thibaut Armel Chérif Gnimadi, Kadio Jean-Jacques Olivier Kadio, Mano Joseph Mathew, Haby Diallo, Abdoul Karim Soumah, Alpha Kabiné Keita, Castro Gbêmêmali Hounmenou, Nicolas Fernandez-Nuñez, Nicole Vidal, Emilande Guichet, Ahidjo Ayouba, Eric Delaporte, Martine Peeters, Abdoulaye Touré, Alpha Kabinet Keita

**Affiliations:** 1Centre de Recherche et de Formation en Infectiologie de Guinée (CERFIG), Université Gamal Abder Nasser de Conakry, Conakry 6629, Guinea; olivier.kadio@cerfig.org (K.J.-J.O.K.); haby.diallo@cerfig.org (H.D.); abdoul.soumah@cerfig.org (A.K.S.); kabine.keita@cerfig.org (A.K.K.); castro.hounmenou@cerfig.org (C.G.H.); abdoulaye.toure@cerfig.org (A.T.); 2EFREI Research Lab, Panthéon Assas University, 30-32 Avenue de la République, 94800 Villejuif, France; mano.mathew@efrei.fr; 3Laboratoire Génomique, Bioinformatique et Chimie Moléculaire, EA7528, Conservatoire National des Arts et Métiers, HESAM Université, 2 Rue Conté, 75003 Paris, France; 4Institut de Recherche pour le Développement (IRD), INSERM, TransVIHMI, University of Montpellier, 34394 Montpellier, France; nicolas.fernandez@ird.fr (N.F.-N.); nicole.vidal@ird.fr (N.V.); emilande.guichet@ird.fr (E.G.); ahidjo.ayouba@ird.fr (A.A.); eric.delaporte@ird.fr (E.D.); martine.peeters@ird.fr (M.P.)

**Keywords:** SARS-CoV-2, genetic diversity, phylodynamic, guinea

## Abstract

In Guinea, genomic surveillance has been established to generate sequences of and to identify locally circulating SARS-CoV-2 variants. This study aims to describe the distributions, genetic diversity, and origins of SARS-CoV-2 lineages circulating in Guinea during the COVID-19 pandemic. A migration analysis was performed by selecting all sequences generated in Guinea for variants of concern and interest. From March 2020 to December 2023, 1038 sequences were generated in Guinea and submitted to the Global Initiative on Sharing All Influenza Data (GISAID) database. Of these, 73.1% corresponded to SARS-CoV-2 variants of concern, which were further grouped into Omicron (69.4%), Delta (21.9%), Alpha (6.6%), and Eta (2.1%). Other variants accounted for 26.9% of the total. Among the total variants analyzed, 75 importations into Guinea from various countries worldwide were identified. Most of the importations (40%) originated from African countries, followed in significance by those from European countries (25.3%) and Asia (18.6%). A significant migratory flow was observed within Guinea. The genomic surveillance reported in this study revealed the diversity of SARS-CoV-2 variants circulating in Guinea, emphasizing the importance of large-scale sequencing analyses in understanding the dynamics of the pandemic.

## 1. Introduction

Severe Acute Respiratory Syndrome coronavirus 2 (SARS-CoV-2) is a highly transmissible RNA virus responsible for the pandemic of Coronavirus Disease 2019 (COVID-19) [1]. This respiratory virus first emerged in Wuhan, China, in December 2019 and is primarily transmitted through respiratory droplets [2]. The rapid spread of SARS-CoV-2 has been attributed to person-to-person contact, the inconsistent use of face masks, inadequate sanitary measures, and limited access to vaccines in many regions [1].

Due to their significant genetic variability and widespread transmission, multiple SARS-CoV-2 variants have emerged, leading to distinct epidemics that have occurred concurrently or successively [3]. The World Health Organization (WHO) categorizes these variants into three main groups: Variants of Concern (VOCs), Variants of Interest (VOIs), and Variants Under Monitoring (VUMs) [4]. These classifications are based on factors such as genetic mutations, transmissibility, disease severity, and ability to evade the immune response elicited by current vaccines, convalescent plasma treatments, or the use of therapeutic monoclonal antibodies [1].

The dissemination of SARS-CoV-2 variants across the globe underscores the importance of understanding both local and global transmission patterns. Such insights are crucial for informing public health policies. While genomic surveillance has been widely implemented globally, the accumulation of SARS-CoV-2 genomic data in western sub-Saharan Africa, including Guinea, has been slow [5].

During the COVID-19 pandemic, several variants exhibited extensive genomic mutations compared to the original Wuhan-01 strain, leading to increased viral infectivity and potential immune escape in humans [3]. As the pandemic has been ongoing for more than three years, our understanding of SARS-CoV-2 and its variants has improved significantly. The clinical and genetic information gathered during this time has been invaluable to global public health [6].

Viral evolution occurs by the gradual accumulation of mutations over generations, allowing adaptation to host environments. However, in some cases, a cluster of mutations can appear simultaneously, resulting in a sudden jump in viral evolution [7]. Omicron, the most divergent VOC to date, exemplifies this phenomenon. It has spread rapidly and unpredictably, in a pattern reminiscent of the outbreak of the Delta variant that emerged in India in 2021 [8,9,10]. Most VOCs and VOIs, including Alpha, Beta, Gamma, Delta, Epsilon, Eta, Iota, Kappa, and Lambda, were first identified across multiple countries in 2020, while the Mu and Omicron variants were detected in 2021 [1]. It is important to note that while viruses constantly evolve, not all mutations result in increased transmissibility or severity. Understanding the mutation patterns of SARS-CoV-2 is crucial for developing effective vaccines and treatments for COVID-19 [11].

In Guinea, as in many subSaharan African countries, four primary waves of COVID-19 epidemics have been observed, with a cumulative total of 38,572 confirmed cases and 468 deaths (Worldometers.info). The first COVID-19 case in Guinea was identified on 12 March 2020 in a traveler from Europe [12]. The origin of this index case was linked to the traveler’s history. Still, the probable origins of the subsequent VOCs or VOIs that circulated during the pandemic remain unexplored.

Data on the origins of these variants or their evolution based on genomic data in Guinea are very limited. This study aimed to fill this gap by performing bioinformatic, phylogenetic, and phylogeographic analyses on large genomic sequences obtained from the international GISAID database. We described the genetic diversity, distributions, and origins of SARS-CoV-2 VOCs and VOIs circulating in Guinea during the COVID-19 pandemic.

## 2. Materials and Methods

### 2.1. Data Collection

This was an analysis of genomic data of SARS-CoV-2 from GISAID between March 2020 and December 2023. To perform the phylogenetic and phylogeography analysis on all variants of concern (VOC) and variants of interest (VOI) identified in Guinea during the active COVID-19 pandemic, we retrieved and downloaded from GISAID the sequence data sets compiled for each lineage by Emma Hodcroft and collaborators on [https://covariants.org/] via Nextstrain, accessed on 21 January 2024: (Alpha (B.1.1.7), Delta (B.1.617.2), Eta (B.1.525), and Omicron {BA.1, BA.2, BA.5, BQ.1, and XBB}) [13]. Each dataset, comprising Guinean sequences and global reference sequences, includes outgroup sequences from earlier lineages that circulated previously. All sequences were downloaded on 17 December 2023. The Guinean sequences were selected and integrated with the reference datasets. We included all Guinean sequences in our analysis, irrespective of the laboratory of origin or the sequencing platform used.

### 2.2. Phylogeography Reconstruction

For each dataset, we retrieved sequences that reflected the sampling period of each VOC (first and last sampling date) in Guinea. We then aligned these sequences against Guinean sequences using Nextalign v2.14.0. The aligned sequences were visualized and edited using AliView v1.28, a fast and lightweight alignment viewer and editor suitable for large datasets [14]. To minimize ambiguities, we masked 100 to 150 base pairs from the beginning and the end of each sequence.

Maximum likelihood trees for each alignment were inferred in IQ-TREE multicore version 2.2.6 COVID-edition using the Ultrafast model selection [15]. All trees were inferred with a general time reversible (GTR) model of nucleotide substitution using empirical base frequencies (+F), a proportion of invariable sites (+I), and a discrete Gamma model with default 4 rate categories (G4).

Time-scaled phylogenetic trees based on sampling dates were then generated using 7.0 × 10^−4^ nucleotide substitutions per site per year with a standard deviation of 3.5 × 10^−4^ using Treetime version 0.11.2, as defined in [16]. We performed molecular-clock testing before the final tree building, and all outliers that deviated more than three interquartile ranges from the root-to-tip regression were removed.

For a phylogenetic representation of the variants, we subsampled each dataset using Augur version 25.4.0 by including a maximum of 500 sequences, and all the sequences from Guinea were retained [17]. All steps of the analysis were repeated to produce a timescaled phylogenetic tree.

### 2.3. Analysis of Introduction of VOI and VOC Variants in Guinea

The migration model was applied to each of the time-tree topologies in Treetime by mapping the country and division (regions of the country) locations of sampled sequences to the external tips of trees. For the division level, we mapped only Guinea sequences with complete metadata (*n* = 1017). The migration model treats locations as discrete traits that evolve through phylogeny [16]. It allowed us to estimate the number of viral transmission events for each VOC between Guinean sequences and the reference sequences. The number of introductions and event dates were estimated using a Python script developed and implemented by Eduan Wilkinson and collaborators [5].

All data analytics were performed using a custom bash script and R scripts, and visualization was performed using the “ggplot” and “ggmap” libraries in RStudio 4.4.1.

### 2.4. Mutations Diversity Analysis

For this analysis, we filtered and retrieved from GISAID all the complete genomes (genomes with more than 29,000 nucleotides) and excluded the low-coverage sequences (>95%, corresponding to 644 sequences) from Guinea. Sequences were uploaded into Nextclade to analyze the mutations among the different strains [18]. We then downloaded the output file that contained the summarized results of the analysis, such as clades, mutations, and quality-control metrics, for statistical analysis. The statistical packages dplyr and tidyr in R were used to summarize and group the data by gene and amino-acid mutation; we then converted all amino-acid mutations below a frequency of 50 to other amino-acid mutations.

## 3. Results

### 3.1. SARS-CoV-2 Variant Distribution

A total of 1038 SARS-CoV-2 genome sequences from Guinea were shared on the GISAID database during the study period. According to the WHO classification, 72% of these sequences correspond to variants of concern (VOCs) and 28% were variants of interest (VOIs), variants under surveillance (VUM), and others. Across all the SARS-CoV-2 genomes, the Omicron variant and its sublineages were the most represented (69.4%), followed by Delta (B.1.617.2) at 21.9%, Alpha (B.1.1.7) at 6.6%, and Eta (B.1.525) at 2.1%.

Different waves of SARS-CoV-2 circulation were observed in Guinea, reflecting a temporal evolution marked by distinct periods of variant dominance. The initial waves were associated with the first cases and the circulation of the ancestral B.1 and B.1.1 lineages, which circulated from the onset of the epidemic in March 2020 until early 2021. During the first half of 2021, co-circulation of multiple lineages was noted, with the Alpha variant being predominant. The third wave was marked by the dominance of the Delta variant, which prevailed until the end of 2021. In December 2021, the first cases of the Omicron variant were detected in Guinea, initiating a new wave dominated by the BA.1 and its sublineages. From March 2022 to the end of 2023, successive waves of Omicron sublineages, including BA.2, BA.5, BQ.1, and XBB.1, were observed. These sublineages led to smaller epidemic waves, reflecting a complex evolutionary and epidemiological dynamic (Figure 1).

Geographically, Guinea is subdivided into eight administrative regions. Of the total sequences generated, 84% (874/1038) were from samples collected in Conakry, the country’s capital city, were many of the administrative departments and the international airport are located. Genomic surveillance also covered the prefectures of Kindia 4.4% (46/1038), Nzérékoré 4.3% (45/1038), Boké 3.9% (42/1038), Mamou 0.9% (9/1038), Kankan 0.5% (5/1038), and Labé 0.3% (3/1038) (Figure 2).

### 3.2. Mutational Analysis

Our analysis showed that most mutations were localized in the spike protein (S) region, followed by ORF1a and the nucleocapsid (N). The most common mutations were D614G, P314L, P681H, T478K, and N501Y (Figure 3).

### 3.3. Origins of Variants of Concern Circulating in Guinea

We carried out a phylogeographic analysis on all the variants of concern (VOCs) that circulated in Guinea to determine the probable origin of the different variants and the dates of introduction of these strains into the country. The variants analyzed were Alpha (B.1.1.7), Delta (B.1.617.2), Eta (B.1.525), Omicron (BA.1, BA.2, BA.5, BQ.1, and XBB), and sublineages. The datasets include sequences for Alpha (2975 genomes, including 50 from Guinea), Delta (3729 genomes, including 108 from Guinea), Eta (3502 genomes, including 26 from Guinea), BA.1 and its sublineages (2303 genomes, including 173 from Guinea), BA.2 and its sublineages (2934 genomes, including 32 from Guinea), BA.5 (1750 genomes, including 44 from Guinea), BQ.1 and its sublineages (1729 genomes, including 108 from Guinea), and XBB.1 and its sublineages (2404 genomes including 106 from Guinea).

We inferred a total of 75 introductions of SARS-CoV-2 variants into Guinea, with the majority originating from African and European countries and accounting for 40% and 25.3%, respectively, and the next-largest group being introductions from Asia, accounting for 18.6% (Figure 4).

For the Alpha variant that emerged in the United Kingdom, we identified seven introductions into Guinea between November 2020 and April 2021; the first importation was from Greece, in Europe, followed by one from Indonesia, in Asia. For the Delta variant, six introductions were inferred between April 2021 and August 2021, including five introductions from an African country, with three (03) from Sierra Leone, which borders Guinea. For the Eta lineage, 12 introduction events were inferred between March 2020 and January 2021; the majority, 4/12, were from Nigeria; the first introduction was from the USA. Regarding the Omicron variant, with its sublineages BA.1, BA.2, BA.5, BQ.1, XBB.1, we inferred a cumulative total of 47 introductions in Guinea; the majority were of BA.2 (16/47) (between December 2021 and September 2022) and originated from India, Côte d’Ivoire, Algeria, Botswana, and other countries; the next-most common were BA.1 (introduced between May 2021 and August 2021), with first introductions from Liberia and Democratic Republic of Congo, and BA.5 (introduced between May 2022 and September 2022); both accounted for 9/47 variants. The BQ.1 variant (8/47) was imported between November 2021 and January 2022 from France, Nigeria, USA, Austria, Senegal, and Germany, and the XBB.1 variant (8/47) was imported between October 2022 and May 2023 from India, Canada, Italy, and Portugal (Figure 5).

Our phylogenetic analysis revealed two distinct patterns for the different variants. We identified dense clusters as well as some scattered sequences for the Alpha, Delta, and Omicron variants (BA.1, BA.5, and XBB.1) and the sublineages. In contrast, for the Eta variant and Omicron sublineages BA.2 and BQ.1, we observed a significant dispersion of sequences throughout the phylogenetic trees. These results support the hypothesis that, in the first case, a single introduction facilitated the rapid local spread of these variants, with a few additional distinct introductions. The second case suggests multiple introductions, possibly accompanied by rapid viral evolution (Figure 6).

The majority of inter-regional introductions within Guinea originated from Conakry, with this region accounting for 67 out of 129 cases (51.9%); the next-most-frequent origins were Nzérékoré, with 24 cases (18.6%), and Coyah and Gueckedou, each contributing 11 cases (8.5%). The first introduction of SARS-CoV-2 was inferred to have reached Conakry from an unidentified source. Subsequent migration movements were observed from Conakry to Coyah (and vice-versa), as well as from Conakry to Gueckedou, Dubreka, and Boffa. Cases were also exported from Nzérékoré to other regions of the country, including Conakry, Kindia, Gueckedou, Forékaria, and Labé (Figure 7).

## 4. Discussion

Throughout the COVID-19 pandemic, Guinea, like other countries, experienced multiple waves of SARS-CoV-2 variants. The first confirmed case in Guinea, detected in March 2020 via PCR test from a European traveler’s sample, prompted health authorities to swiftly implement preventive measures to curb the spread of the virus. Laboratories across the country were equipped with molecular diagnostic tools to enhance the detection of suspected cases. In the initial stages of the pandemic, Guinea lacked local sequencing capacity, which limited its ability to monitor the genetic evolution of circulating SARS-CoV-2 variants. As a result, the first viral genomes were sequenced abroad in collaboration with partner laboratories. This international cooperation allowed for the initial characterization of the strains circulating in the country.

A year after the pandemic began, Guinea successfully developed its local sequencing capacities due to the establishment of genomic surveillance networks, including the Afroscreen network, a French response program against COVID-19, to strengthen the monitoring of the evolution of variants in 13 African countries [19]. This marked a significant advance in the country’s ability to monitor and respond to emerging SARS-CoV-2 variants. The implementation of local sequencing enabled a more comprehensive understanding of the viral strains present within the country, contributing to global efforts to track the spatio-temporal evolution of SARS-CoV-2.

In this study, we investigated the genetic diversity and the distribution of SARS-CoV-2 strains circulating in Guinea by analyzing viral sequences shared in public databases. Between March 2020 and December 2023, Guinea experienced four major epidemic waves, each marked by the circulation of distinct strains or variants.

The first strains identified in March 2020 belonged to the B.1 and B.1.1 lineages, which were classified under Clades 20A and 20B according to the Nextstrain classification. These early strains evolved into various sublineages, which persisted from 2020 through the first quarter of 2021. According to the data shared by WHO and the Guinea National Agency for Health Security (ANSS), a cumulative 14,532 (37.6%) cases, with 82 (17.5%) deaths, occurred during this period (Appendix A).

In January 2021, Guinea recorded its first cases of Variants of Concern (VOCs), including B.1.1.7 (Alpha), B.1.525 (Eta), and B.1.617.2 (Delta), all falling within Clade 20I. These VOCs were first detected in Guinea between January and May 2021, contributing to the second and third waves of the pandemic in the country [12,20]. The Delta variant dominated the third wave, causing a greater number of deaths, 219/468 (46.7%). The authors suggest that this pattern could be associated with the declaration of the end of the Ebola epidemic in June 2021 and the low vaccination rate of the population, regardless of possible newly emerged mutations or the introduction of the Delta variant [21].

In December 2021, the Omicron variant (BA.1, Clade 21K) was identified in Guinea, marking the beginning of the fourth wave of the pandemic. The Omicron variant, along with its rapidly evolving sublineages, became the dominant strain and persisted throughout the remainder of the pandemic until December 2023, as reflected in sequences shared in public databases.

Most of the SARS-CoV-2 genomes were derived from positive samples collected in Conakry, the capital of Guinea. The presence of the country’s largest international airport in Conakry makes it the site of significant international and subregional air traffic, which contributed to the early and continuous introduction of new variants. We noticed that a limited amount of genomic data was obtained from other regions in Guinea, which may be due to the lack of sequencing capacity in laboratories outside of Conakry. Also, at the beginning of the pandemic, epidemiological surveillance and diagnostic laboratory capacities were established in the capital, leaving other regions underrepresented in genomic surveillance efforts. This centralization of resources further explains why most sequences were obtained from Conakry patients, limiting our understanding of the spread and diversity of SARS-CoV-2 in other areas of the country.

In our analysis, we identified the key mutations present in SARS-CoV-2 genomes from Guinea, focusing on those primarily associated with the diversity of circulating variants. The most frequently observed mutation was D614G, which was extensively reported in numerous studies as the dominant mutation throughout the COVID-19 pandemic and which appears in most SARS-CoV-2 lineages [22]. This mutation, first identified early in the pandemic, enhances viral replication in human lung epithelial cells and primary respiratory tissues by increasing virion infectivity and stability [23,24].

The second-most-frequent mutation observed was P681H, one of the eight mutations identified in the spike protein of the Alpha variant (B.1.1.7), which emerged at the end of 2020. This mutation is believed to improve furin cleavage, facilitating viral entry into host cells [25]. Additionally, the N501Y mutation, which was also detected in our data, is associated with increased transmissibility of the Alpha variant [26].

Another notable mutation, T478K, was identified with high frequency and is characteristic of the Delta variant (B.1.617.2). In early sequence analyses, this mutation was described as unique to Delta, contributing to its enhanced spread and impact during the pandemic [27].

SARS-CoV-2 variants were introduced into Guinea through international and subregional transmission, with individuals travelling from infected countries importing new strains. In addition to these imported cases, the virus spread significantly locally. A migration analysis revealed that most introductions came from African countries, Europe, and Asia.

Similarly, research in the neighboring country of Côte d’Ivoire indicated that many introductions originated from other subregional countries, highlighting the impact of regional transmission [28]. Our analysis suggests that 25.3% of cases in Guinea were introduced from Europe. Wilkinson et al., in their study, observed that 64% of detectable viral introductions in Africa originated from Europe, underscoring the strong epidemiological connection between Europe and Africa [5].

The most notable introductions within Guinea from African countries were those of the Eta variant from Nigeria and the Delta variant from Sierra Leone, a neighboring country. For the Alpha and Delta variants, local transmission was significant, a pattern likely influenced by the closure of borders and the suspension of international flights. Additionally, containment measures implemented during this period may have further limited international introductions while amplifying local spread and leading to the occurrence of the highest death rate in this period.

A significant migratory flow was observed within Guinea, demonstrating that despite the public health measures and controls implemented to limit the circulation of the virus from Conakry to other country regions, its transmission to those regions was not prevented. In addition, many exportations from the Nzérékoré region to different regions of the country were identified, including exportations to Conakry. This could be explained by the potential introduction of strains to Nzérékoré from bordering countries, as the region borders three subregional African countries: Cote d’Ivoire, Sierra Leone, and Liberia.

To our knowledge, this study is the first in Guinea to explore the diversity and phylodynamics of SARS-CoV-2 variants within the country. While it highlights the profiles of the different mutations, the probable origins of strains circulating in Guinea, and the migratory flow between different regions, it also has certain limitations. Specifically, the reference sequences selected may not fully represent the West African sub-region or other areas due to the disproportionately higher number of sequences generated by developed countries with more extensive sequencing resources. In addition to analyzing the introduction and migration of SARS-CoV-2 variants in Guinea, future research should focus on the dynamics of local spread and the evolution of virus mutations. Identifying emerging local mutations and understanding their impact on transmission and pathogenicity could provide valuable information on population susceptibility and inform strategies for preparing for future pandemics.

## 5. Conclusions

This analysis of a large dataset of SARS-CoV-2 sequences, including genomes produced in Guinea during the COVID-19 pandemic, has provided valuable insights into the distribution of variants and mutations circulating in the country. The genetic diversity observed in Guinea closely mirrors patterns seen in many other countries worldwide. Our findings revealed multiple introductions of Variants of Concern (VOCs) into Guinea, with these events directly associated with the various epidemic peaks observed, underscoring the significant impact of intercontinental and inter-regional travel on the spread of the virus.

The importations analysis may have some limitations due to the low representation of sequences from some African countries, particularly from the West African sub-region. This highlights the need to strengthen genomic surveillance in low- and middle-income countries and ensure that it covers samples from the whole country for future epidemics and pandemics. Increased surveillance efforts will facilitate the early detection of emerging mutations and variants and improve our understanding of epidemics, underscoring the essential role of sequencing in the fight against epidemics.

## Figures and Tables

**Figure 1 viruses-17-00204-f001:**
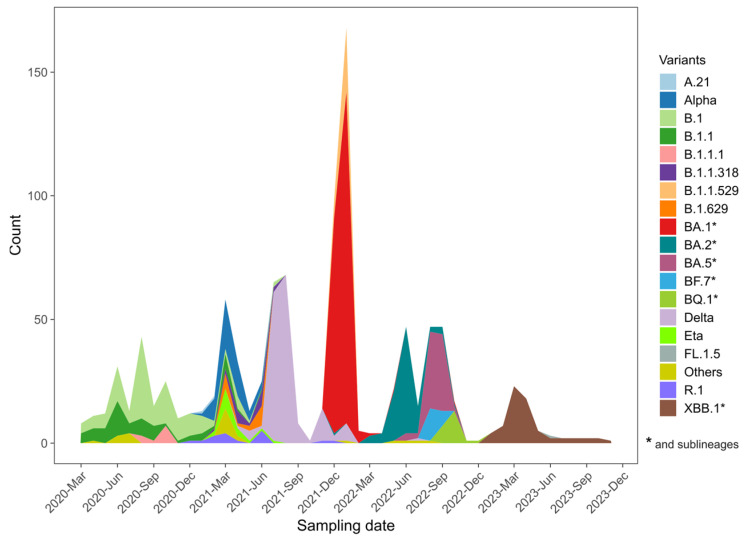
Trends in the prevalence of major variants circulating in Guinea from March 2020 to December 2024. The Y-axis shows the distribution (*n* = 1038) of various variants across the various months (X-axis), while different colors represent the lineages.

**Figure 2 viruses-17-00204-f002:**
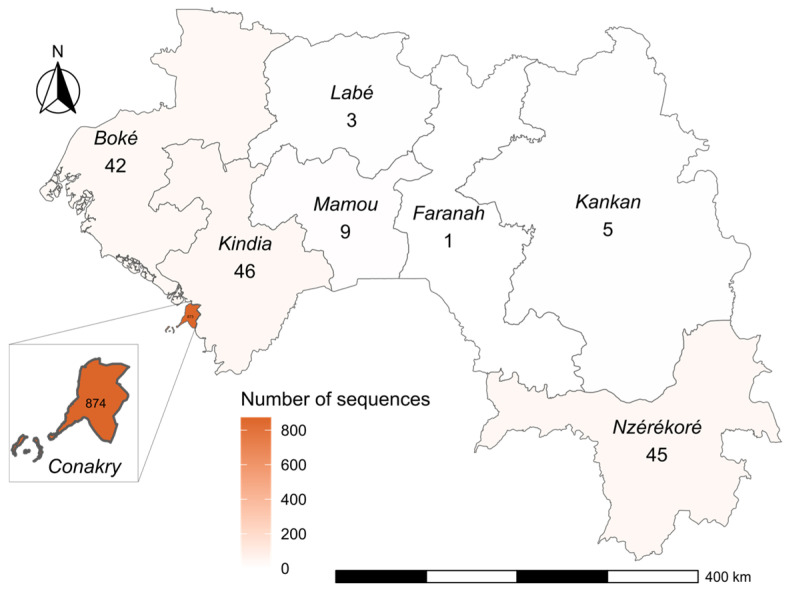
Map showing the sampled regions (*n* = 1038) and the color scheme showing the number of sequences generated from samples collected in each region.

**Figure 3 viruses-17-00204-f003:**
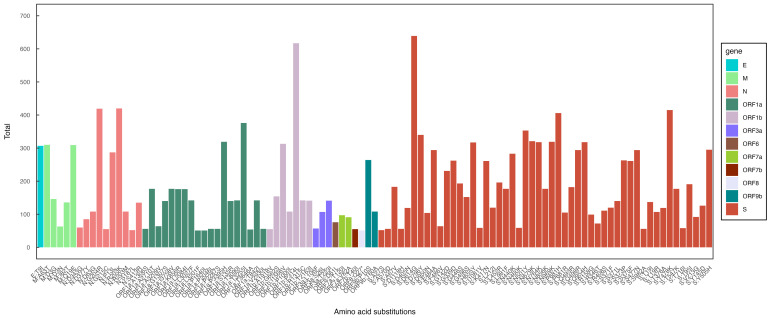
Frequencies of amino-acid substitutions across all the SARS-CoV-2 proteins in all high-quality genomes submitted to GISAID for Guinea (*n* = 644/1038). The mutations are sorted and colored by gene.

**Figure 4 viruses-17-00204-f004:**
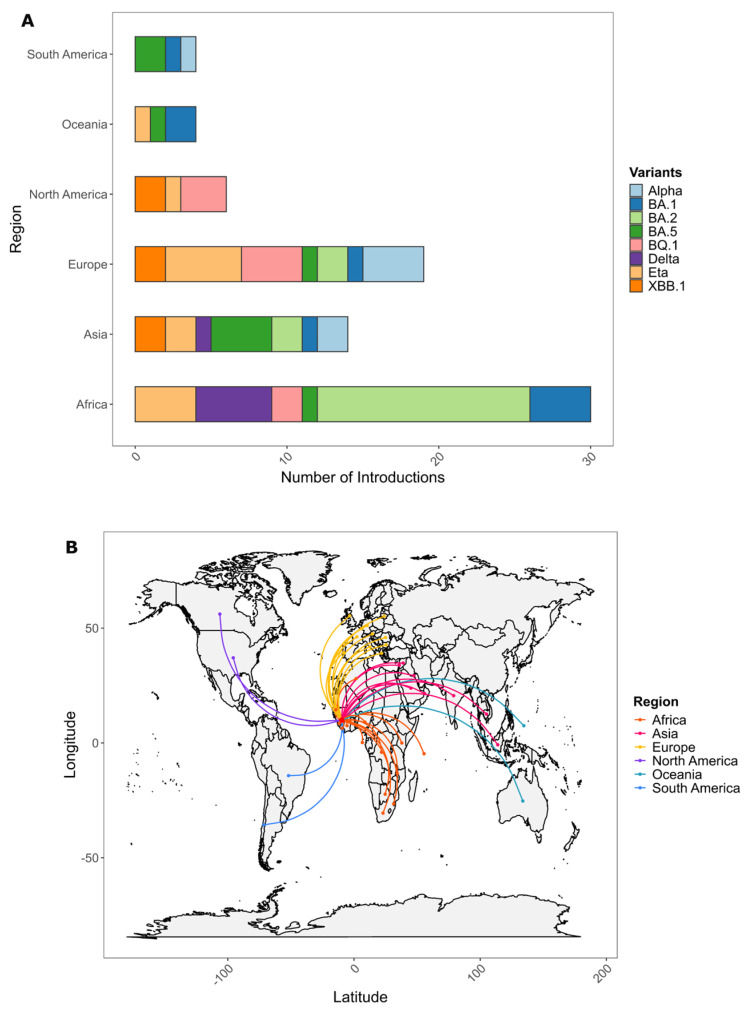
(**A**) Number of importation events by world region of severe acute respiratory syndrome coronavirus 2 (SARS-CoV-2) lineages (Alpha, Eta, Delta, Omicron {BA.1, BA.2, BA.5, BQ.1, XBB.1}, and sublineages) into Guinea from March 2020 to December 2023. (**B**) Geographical distribution of the importations.

**Figure 5 viruses-17-00204-f005:**
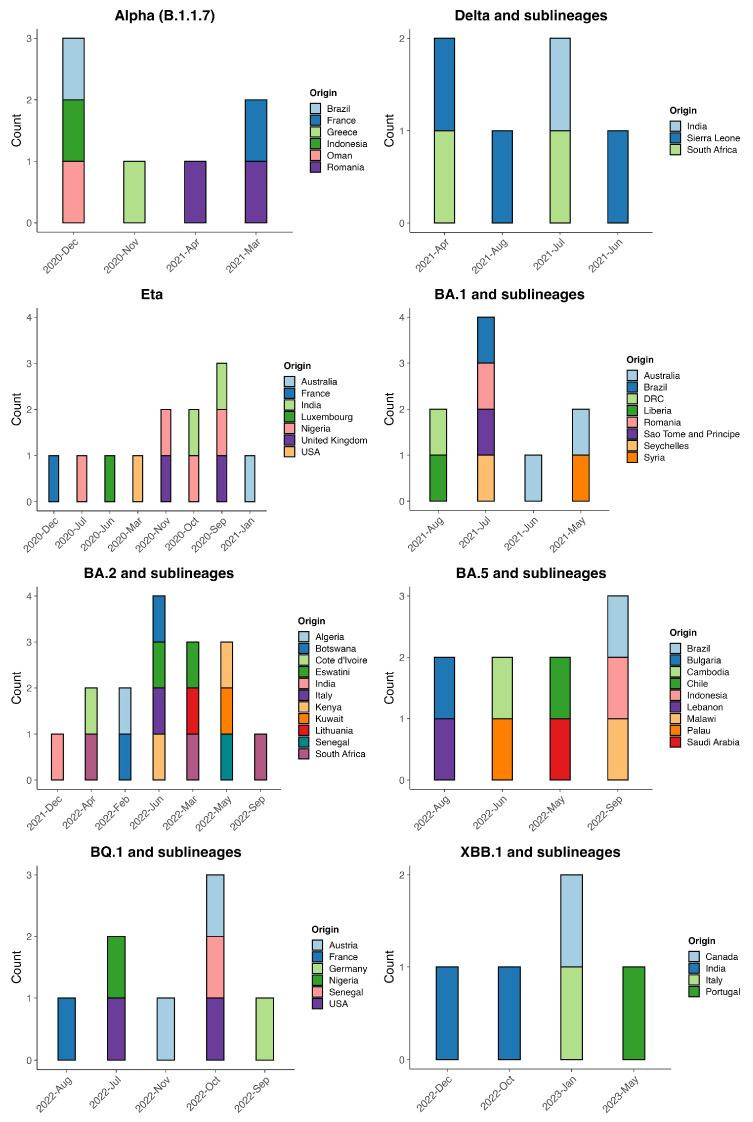
Number and origin of importation events of the different SARS-CoV-2 lineages (Alpha, Eta, Delta, Omicron {BA.1, BA.2, BA.5, BQ.1 and XBB.1.5}, and sublineages) into Guinea between March 2020 and December 2023.

**Figure 6 viruses-17-00204-f006:**
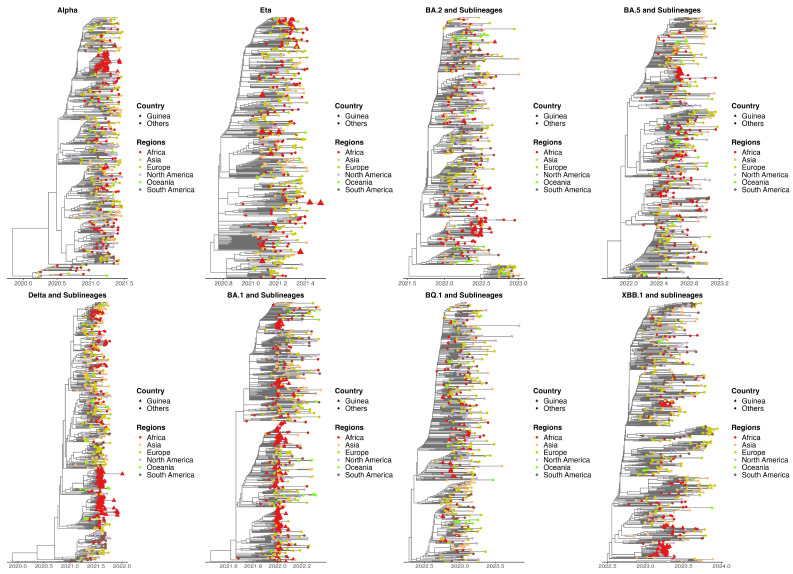
Timescale phylogeny of SARS-CoV-2 lineages (Alpha, Eta, Delta, Omicron {BA.1, BA.2, BA.5, BQ.1 and XBB.1.5} and sublineages). Five hundred sequences were subsampled from a global dataset to maximize genetic distances while retaining all genomes from Guinea. The branches are scaled in decimal time, and sampling dates are capped on each lineage’s last sampling date, the latest sampling month in Guinea in this study.

**Figure 7 viruses-17-00204-f007:**
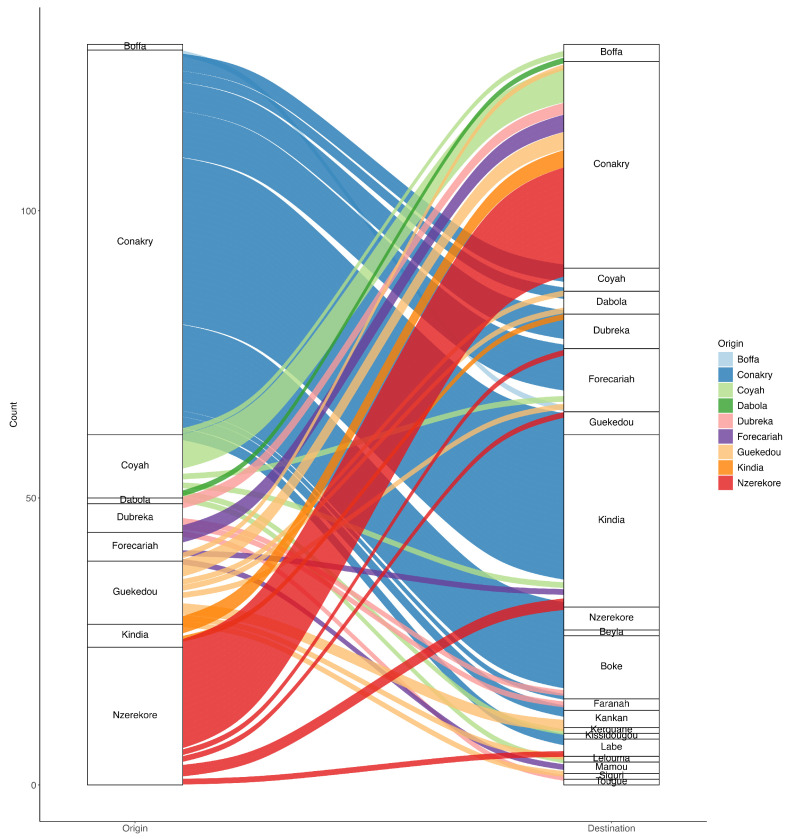
Number of SARS-CoV-2 imports and exports into and out of various regions in Guinea.

## Data Availability

All genome sequences and associated metadata in this dataset are published in GISAID’s EpiCoV database. To view the contributors of each sequence with details such as accession number, Virus name, Collection date, Originating Lab and Submitting Lab, and the list of Authors, visit https://doi.org/10.55876/gis8.241101td accessed on 1 November 2024. All the codes and data used for the analysis reported in this study are publicly available at https://github.com/armel001/Guinea-SARS-CoV2-Spatiotemporal accessed on 1 November 2024.

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
