# Peer review of "Genetic Diversity and Spatiotemporal Distribution of SARS-CoV-2 Variants in Guinea: A Meta-Analysis of Sequence Data (2020–2023)"

_viruses, 2025, doi:10.3390/v17020204_

Round 1

Reviewer 1 Report

Comments and Suggestions for Authors

The manuscript by Gnimadi et al. analyzes SARS-CoV-2 variant data from Guinea between March 2020 and December 2023. Guinea experienced four major epidemic waves, each dominated by different variants, which is similar to other countries. This study relies heavily on genomic data obtained from GISAID. Without clinical data (e.g., disease severity, hospitalization rates), it is hard to comprehensively understand the impact of different variants on human health. The results of this manuscript were mainly compatible with earlier research conducted in other nations. This paper has low originality/novelty; I don't notice any new information added in this field. This is the major drawback of this manuscript.

Specific comments:

1. COVID-19 pandemic has passed for years, what new information did this retrospect study add into this filed?

2.  The abbreviation "GISAID" first appears in line 27 without the entire name. Please correct this.

3.  Please enlarge words in figure 4 and replace figure by a higher resolution one.

Author Response

Thank you very much for taking the time to review this manuscript. We are grateful for these insightful comments. Please find the detailed responses below. We have highlighted the changes within the re-submitted manuscript.

Comments 1: COVID-19 pandemic has passed for years; what new information did this retrospective study add into this field?

Response 1: Thank you for your valuable comment. We acknowledge that the peak of the COVID-19 pandemic has passed; however, our retrospective study provides important insights that contribute to the ongoing understanding of the pandemic dynamics, particularly in resource-limited settings like Guinea. We think that this study is the first in Guinea to analyze both the local geographical distribution of SARS-CoV-2 variants, importation cases and the internal migratory flow of the virus across country regions. Our findings reveal the gaps in the effectiveness of the public health measures implemented to prevent and control the virus’s spread within the country. It's also important to note that although the WHO has declared the epidemic over, some countries and research centres are still conducting sentinel surveillance for SARS-Cov-2. It is, therefore, essential that genomic data from the acute phase and beyond are sufficiently analyzed.

Comments 2: The abbreviation "GISAID" first appears in line 27 without the entire name. Please correct this.

Response 2: Thank you for pointing this out. We agree with this comment, and this has been considered on Page 1 / line 27 [Global Initiative on Sharing All Influenza Data (GISAID)]

Comments 3: Please enlarge words in Figure 4 and replace the figure with a higher resolution one.

Response 3: This comment has been taken into account.

Reviewer 2 Report

Comments and Suggestions for Authors

Authors conducted a meta-analysis of a total of 1038 SARS-CoV-2 sequences generated in Guinea and submitted to the GISAID database and reported demographic distribution and circulation of variants. A "Migration model" was used to predict introduction of variants from other countries. However, the introduction of variants cannot be differentiated from the variant evolution within the Guinea without contact-tracing data from local registry. Thus, the "Evolution" and "An Analysis" in the article title are misleading and should be changed to "distribution" and "A Meta-analysis", respectively. It is also not clear what "Background sequence" in the Materials and Methods section was used as the reference for aligning other sequences. Although the migration of variants is important, the local spread and mutation evolution should also be examined to determine the COVID-19 susceptibility and SARS-CoV-2 mutation rate within Guinea, which would provide more valuable biological data for next pandemic.     

Comments on the Quality of English Language

Grammatical and typographic errors are all over the manuscript and should be corrected.

Author Response

Thank you very much for taking the time to review this manuscript. We are grateful for these insightful comments. Please find the detailed responses below. We have highlighted the changes within the re-submitted manuscript.

Comment 1: Thus, the "Evolution" and "An Analysis" in the article title are misleading and should be changed to "distribution" and "A Meta-analysis", respectively.

Response 1: Thank you for pointing this out. We agree with the comment and consider it in the Title Page 1 / Line 2 and 3

Comment 2: It is also not clear what "Background sequence" in the Materials and Methods section was used as the reference for aligning other sequences.

Response 2: Thank you for this comment. We have taken this into account and used it across the Materials and Methods “reference sequence” instead of “Background sequence” for better understanding. The “Reference sequences” represent world sequences selected for each variant that we aligned against Guinean sequences for the phylogenetic and phylogeography analysis

Comment 3: Although the migration of variants is important, the local spread and mutation evolution should also be examined to determine the COVID-19 susceptibility and SARS-CoV-2 mutation rate within Guinea, which would provide more valuable biological data for next pandemic.

Response 3: Thank you for pointing this out. We agree with this comment. We acknowledge that our study primarily focused on the introduction and migration of variants and did not explore the local transmission dynamics and evolutionary patterns. This is indeed a limitation of our work. In the revised manuscript, we have highlighted this limitation and the need for future studies. Line 367-372 / Page 9

Additional clarifications

In addition to the above comments, we carefully reviewed the entire manuscript and corrected all grammatical and typographic errors by the reviewers

We also provided two supplementary tables with epidemiological data on the number of confirmed cases and hospital deaths during the major waves, along with the dominant SARS-CoV-2 variants. This is in order to provide a clearer understanding of the impact of the different waves on public health in Guinea. The additional comments have also been highlighted in the Discussion section

Round 2

Reviewer 1 Report

Comments and Suggestions for Authors

No further comments.

Author Response

Thank you again for taking the time to review this manuscript. We appreciate your interest in this work

Reviewer 2 Report

Comments and Suggestions for Authors

The revised manuscript may be acceptable, pending the extensive correction for grammatical errors and readability of English. This Reviewer took the liberty of improving the manuscript to provide an exemplified revision (in red strikethrough and/or red text) in the Abstract in the attached file. 

Comments on the Quality of English Language

Authors are encouraged to reach out to English-editing service or English-speaking colleague for revision in other sections.      

Author Response

Thank you again for taking the time to review this manuscript. We appreciate your interest in this work and comments on improving its quality. Please find below the detailed responses

Comments 1: Authors are encouraged to reach out to English-editing service or English-speaking colleague for revision in other sections.      

Response 1: Thank you for pointing this out. We have taken this into account.